# Autosomal Dominant Alzheimer’s Disease Mutations in Human Microglia Are Not Sufficient to Trigger Amyloid Pathology in WT Mice but Might Affect Pathology in 5XFAD Mice

**DOI:** 10.3390/ijms25052565

**Published:** 2024-02-22

**Authors:** Carmen Romero-Molina, Sarah M. Neuner, Marcelina Ryszawiec, Alice Pébay, Edoardo Marcora, Alison Goate

**Affiliations:** 1Ronald M. Loeb Center for Alzheimer’s Disease, Department of Genetics and Genomic Sciences, Icahn School of Medicine at Mount Sinai, 1 Gustave L. Levy Place, New York, NY 10029, USA; carmen.romeromolina@mssm.edu (C.R.-M.); neuner.sarah@gmail.com (S.M.N.); marcelina.ryszawiec@mssm.edu (M.R.); edoardo.marcora@mssm.edu (E.M.); 2Department of Anatomy and Physiology, Department of Surgery, Royal Melbourne Hospital, The University of Melbourne, Melbourne, VIC 3010, Australia; apebay@unimelb.edu.au; 3Department of Neurology, Washington University, St. Louis, MO 63110, USA

**Keywords:** Alzheimer’s disease, microglia, amyloid

## Abstract

Several genetic variants that affect microglia function have been identified as risk factors for Alzheimer’s Disease (AD), supporting the importance of this cell type in disease progression. However, the effect of autosomal dominant mutations in the amyloid precursor protein (*APP*) or the presenilin (*PSEN1/2*) genes has not been addressed in microglia in vivo. We xenotransplanted human microglia derived from non-carriers and carriers of autosomal dominant AD (ADAD)-causing mutations in the brain of hCSF1 WT or 5XFAD mice. We observed that ADAD mutations in microglia are not sufficient to trigger amyloid pathology in WT mice. In 5XFAD mice, we observed a non-statistically significant increase in amyloid plaque volume and number of dystrophic neurites, coupled with a reduction in plaque-associated microglia in the brain of mice xenotransplanted with ADAD human microglia compared to mice xenotransplanted with non-ADAD microglia. In addition, we observed a non-statistically significant impairment in working and contextual memory in 5XFAD mice xenotransplanted with ADAD microglia compared to those xenotransplanted with non-ADAD-carrier microglia. We conclude that, although not sufficient to initiate amyloid pathology in the healthy brain, mutations in *APP* and *PSEN1* in human microglia might cause mild changes in pathological and cognitive outcomes in 5XFAD mice in a manner consistent with increased AD risk.

## 1. Introduction

Alzheimer’s disease (AD) is a progressive neurodegenerative disorder and the most common cause of dementia. AD is associated with the abnormal accumulation of amyloid plaques and neurofibrillary tangles in the brain, coupled with neurodegeneration, gliosis, lipidosis, and vascular alterations. Although most AD cases occur sporadically in people over the age of 65, heterozygous germline mutations in either the amyloid precursor protein *(APP)* gene or the presenilin *(PSEN1* and *PSEN2)* genes cause early-onset, autosomal-dominant AD (ADAD). APP is a transmembrane protein that is processed to form amyloid-beta (Aβ) peptides, which aggregate to form amyloid plaques. PSEN1 and PSEN2 form the catalytic subunit of the γ-secretase enzyme that cleaves APP to form Aβ peptides. The genetic link between amyloidogenic APP processing and AD led to the emergence of the amyloid cascade hypothesis (ACH) [1], which posits that amyloid aggregation and deposition is the triggering event in AD pathogenesis and has led to the development of AD treatments targeting Aβ [2,3,4].

Microglia, the resident macrophages of the central nervous system, regulate brain development, maintain neuronal networks, and respond to brain infection and injury to bring about immune response and tissue repair. They are responsible for the elimination of microbes, dead cells, redundant synapses, and protein aggregates. In animal models of amyloid pathology, it has been suggested that microglia attempt to clear/compact amyloid plaques [5,6,7], but their activity can become dysregulated. Furthermore, microglia can also be activated by other pathological changes in the brain, such as neurofibrillary tangles [8,9] and oxidative stress [10]. Recent research has highlighted the complexity of microglia in AD, revealing different subsets of microglia with distinct gene expression profiles [11]. Several genetic variants in genes and gene regulatory elements that affect microglial function have been identified as risk factors for AD, confirming the importance of this cell type in AD etiology [12]. Notably, loss of TREM2 function (*Trem2*−/−) has little effect on plaque burden but does affect the number of microglia surrounding plaques and the structure/compaction of the plaques [13]. *APP* and *PSEN1* are broadly expressed in all cell types in the brain and have been implicated in the inflammatory process [14,15] by modulating the release of pro-inflammatory cytokines [16,17], cell motility [18], Ca^2+^ signaling and phagocytosis [19], suggesting that these genes may play an important yet underappreciated role in microglia.

Understanding the complex interplay between microglia and AD pathology is crucial for developing effective therapies to treat this devastating disease. Xenotransplantation of human microglia into the mouse brain constitutes an innovative system to evaluate the effects of human genetic variants associated with AD in vivo. High similarity has been shown between xenotransplanted human microglia and human ex vivo microglia from patients at the transcriptomic level [20]. In this work, we used induced pluripotent stem cell (iPSC) lines derived from carriers and non-carriers of ADAD-causing mutations in *APP* and *PSEN1 (APP V717L, APP V717I, PSEN1 A79V, PSEN1 G217R)*. For *APP V717I* and for *PSEN1 A79V* and *G217R* mutations, we used non-carrier donor lines from members of the same family. These lines were derived from individuals enrolled in the Dominantly Inherited Alzheimer Network (https://dian.wustl.edu/, accessed on 25 January 2024). Each mutation is associated with autosomal dominantly inherited AD, although *PSEN1 A79V* shows more variable penetrance than the other mutations in other cohorts [21,22]. Enzymological analysis in vitro has shown that each of these mutations affects ɣ-secretase cleavage of APP [23], leading to reduced processivity and enhanced production of longer and more aggregation-prone Aβ peptides [24]. We xenotransplanted human hematopoietic progenitors from these lines into the brain of immune-deficient hCSF1 WT or 5XFAD mice, where they spontaneously differentiated into microglial cells. hCSF1 mice express human mCSF (*CSF1*) on a *Rag2/Il2rg* deficient background, significantly increasing their ability to engraft and maintain human hematopoietic cells. hCSF1-5XFAD (shortened to 5XFAD from now on) mice overexpress co-integrated *APP* and *PSEN1* transgenes containing five familial AD mutations (the Swedish (*K670N/M671L*), Florida (*I716V*), and London (*V717I*) mutations in *APP*, and the *M146L* and *L286V* mutations in *PSEN1*) and recapitulate aspects of human AD, including age-dependent cognitive deficits, amyloid accumulation and neuroinflammation [25]. We tested whether ADAD-causing mutations in human microglia are sufficient to cause amyloid pathology in WT mice and evaluated whether these mutations modify amyloid and neuritic pathology, microglia interaction with amyloid plaques, astrogliosis, and cognitive outcomes in vivo in a mouse model of AD (5XFAD).

## 2. Results

### 2.1. WT and ADAD Human Microglia Successfully Engraft into the Mouse Brain

This work confirms the feasibility of xenotransplantation of both wild-type (WT) and mutation-carrying human microglia in the mouse brain, a cutting-edge technique that will allow investigation into the impact(s) of disease-relevant mutations in human cells in a more biologically relevant setting. Hematopoietic progenitor cells (HPCs) were differentiated in vitro from iPSC lines derived from four carriers of *APP* (*V717L* and *V717I*) or *PSEN1* (*A79V* and *G217R*) ADAD-causing mutations and four non-carriers (Table 1). HPCs were injected into the hippocampus and cortex of neonatal (P2) WT and 5XFAD hCSF1 mice according to an established protocol [20]. Behavior tests were performed at 3 and 5–6 months of age and mice were sacrificed at 6 months for immunohistochemistry analyses (Figure 1A). Human xenotransplanted microglia were identified using a combination of the pan-microglia marker IBA1 and the human-specific nuclear marker Ku80. Double-positive microglia, considered to be mature human microglia, were largely identified in target areas of the hippocampus and cortex (Figure 1B). Thus, for all subsequent analyses, we selected these regions as our regions of interest. Total microglia (IBA1+) numbers in WT or 5XFAD mice did not differ regardless of the human cell line used for xenotransplantation (Figure 1C, ADAD − WT (95% confidence interval), in WT mice = 1375 (−1589, 4338), *p*-value = 0.421; in 5XFAD mice = 1076 (−1774, 2486), *p*-value = 0.745; N=4 WT + 4 ADAD donor lines, 1–5 xenotransplanted mice per donor line, 4–7 brain sections analyzed per mouse). Across mice, an average of 42% of all microglia were found to be of human origin due to positive nuclear Ku80+ staining, although high variability was observed (±17%). In WT mice, human microglia derived from iPSC lines with ADAD mutations trended toward lower engraftment efficiency, although this difference was not statistically significant (ADAD − WT (95% confidence interval = −15.5 (−32.8, 1.88) %, *p*-value = 0.151, N=4 WT + 4 ADAD donor lines, 1–5 xenotransplanted mice per donor line, 4–7 brain sections analyzed per mouse). This difference was not observed in 5XFAD mice (ADAD − WT (95% confidence interval) = −3.89 (−25, 17.2) %, *p*-value = 0.727, N=4 WT + 4 ADAD donor lines, 1–5 xenotransplanted mice per donor line, 4–7 brain sections analyzed per mouse).

### 2.2. ADAD Mutations in Human Microglia Are Not Sufficient to Trigger Amyloid Pathology in WT Mice

Extensive research has explored the impact of *APP* and *PSEN1* mutations in neurons, as transgenic mice have largely been engineered to exclusively express mutant human transgenes under the control of neuron-specific promoters, such as the Thy1 promoter in the case of the traditional 5XFAD model. These models have clearly shown human Aβ accumulation and amyloid plaque formation, demonstrating that neuronal expression of only mutant *APP* [26,27] or both mutant *APP* and *PSEN1* [25,28] are sufficient to cause these neuropathologies. However, the extent to which microglia-derived mutant *APP* and/or *PSEN1* contribute to these phenotypes is poorly understood. To address this question, we xenotransplanted WT mice with ADAD human microglia in order to specifically control the source of mutant *APP* or *PSEN1*. As shown in Appendix A, when brain sections from WT mice xenotransplanted with ADAD human microglia were immunostained using an antibody against human Aβ42, which recognizes endogenous levels of total Aβ (including Aβ37, Aβ38, Aβ39, Aβ40, and Aβ42), we did not detect any positive staining of amyloid deposits. Therefore, we can conclude that human microglia carrying either *APP* or *PSEN1* ADAD mutations in isolation are not sufficient to cause amyloid pathology in the mouse brain.

### 2.3. Amyloid Pathology in 5XFAD Mice Xenotransplanted with WT or ADAD Human Microglia

Next, we focused on 5XFAD mice to investigate whether human microglia with ADAD mutations were sufficient to exacerbate the amyloid pathology typically observed in 5XFAD mice. The 5XFAD transgene is considered one of the more aggressive amyloid models available, with mice typically developing amyloid pathology at 3–4 months old depending on genetic background [29]. To allow sufficient time for amyloid pathology to develop (https://www.alzforum.org/research-models/5xfad-c57bl6, accessed on 10 January 2024), we sacrificed mice at 6 months of age and performed IHC analyses to investigate plaque load and morphology. First, we stained the brain sections with the same Aβ42 antibody used to investigate WT brain sections described earlier, but we did not observe differences between 5XFAD mice xenotransplanted with ADAD or WT microglia as quantified by either the percentage of total area covered by Aβ42 staining (ADAD/WT (95% confidence interval) = 1.21 (0.45, 3.23), *p*-value = 0.717) or by the intensity of amyloid staining (ADAD/WT (95% confidence interval) = 0.69 (0.43, 3.51), *p*-value = 0.704), (N = 4 WT + 4 ADAD donor lines, 1–5 xenotransplanted mice per donor line, 4–7 brain sections analyzed per mouse) (Figure 2A).

Using the Imaris software (version 10.0) for 3D reconstruction, we performed more detailed analyses of plaque volume and morphology (Figure 2B, N = 3 WT + 4 ADAD donor lines, 1–3 xenotransplanted mice per donor line, 3–4 brain sections analyzed per mouse, 5 plaques quantified per section). We observed a non-statistically significant increase in plaque volume in the brain of 5XFAD mice xenotransplanted with ADAD human microglia compared to 5XFAD mice xenotransplanted with WT human microglia (ADAD/WT (95% confidence interval) = 1.33 (0.77, 2.29), *p*-value = 0.319). In addition, we observed a decrease in plaque sphericity, which may reflect a reduction in plaque compaction (ADAD − WT (95% confidence interval) = −0.039 (−0.096, 0.017), *p*-value = 0.196).

To further explore plaque morphology, we stained for Thioflavin S (ThioS), reported to specifically mark dense-core plaques (Figure 2C). No changes were observed in ThioS as quantified by either the percent area covered by ThioS staining (graph not shown, ADAD/WT (95% confidence interval) = 1.07 (0.27, 4.27), *p*-value = 0.930, N=3 WT + 4 ADAD donor lines, 1–3 xenotransplanted mice per donor line, 3–7 brain sections analyzed per mouse) or by the intensity of ThioS staining (ADAD/WT (95% confidence interval) = 1.07 (0.26, 4.4), *p*-value = 0.925, N = 3 WT + 4 ADAD donor lines, 1–3 xenotransplanted mice per donor line, 3–7 brain sections analyzed per mouse). However, we found a non-statistically significant decrease in ThioS average plaque size in the presence of human microglia carrying ADAD mutations (ADAD/WT (95% confidence interval) = 0.82 (0.46, 1.46), *p*-value = 0.519; N = 3 WT + 4 ADAD donor lines, 1–3 xenotransplanted mice per donor line, 3–7 brain sections analyzed per mouse). As ThioS specifically stains for dense-core plaques, this finding suggests a reduction in the plaque core. Together with the Aβ42 results, where we observed an increase in total plaque volume (Figure 2B), our results suggest a reduction in plaque compaction in 5XFAD mice xenotransplanted with ADAD human microglia as compared to WT human microglia. There were no changes in the number of plaques either with Aβ42 or with ThioS staining (Appendix A).

### 2.4. Microglia–Plaque Interaction in AD Mice Xenotransplanted with WT or ADAD Human Microglia

It has been reported that mutations in other AD risk genes, such as TREM2 [30] or PLCG2 [31], modify microglia–plaque interaction in 5XFAD mice. We stained brain sections using IBA1 and Ku80 antibodies as described above and counted the number of microglial cells in direct contact with amyloid plaques, normalized by the plaque volume, in the cortical region (Figure 3A). All xenotransplanted 6-month-old 5XFAD mice exhibited robust microglial clustering around plaques. We found a non-statistically significant decrease both in the number of total (IBA1+, Figure 3B, ADAD/WT (95% confidence interval) = 0.80 (0.50, 1.31), *p*-value = 0.397) as well as human (IBA1+/Ku80+, Figure 3C, ADAD/WT (95% confidence interval) = 0.30 (−0.44, 1.03), *p*-value = 0.427) microglia clustered around amyloid plaques in 5XFAD mice xenotransplanted with ADAD human microglia as compared to 5XFAD mice xenotransplanted with WT human microglia (N = 3 WT + 4 ADAD donor lines, 1–3 xenotransplanted mice per donor line, 3 brain sections analyzed per mouse, 5 plaques quantified per section). Using 3D reconstruction in Imaris, we also quantified the volume covered by microglia within 5 μm of an amyloid plaque. We found a non-statistically significant decrease in 5XFAD xenotransplanted with ADAD human microglia compared to WT human microglia (Figure 3D, ADAD/ WT (95% confidence interval) = 0.66 (0.38, 1.14) %, *p*-value = 0.179).

In order to address the mechanisms underlying a reduction in plaque-associated microglia, we evaluated the motility of microglia differentiated from these same HPCs in vitro. As shown in Figure 3E, and in accordance with previously published data [18], we did not observe a reduction in mobility in ADAD microglia compared to WT microglia (ADAD − WT (95% confidence interval) = 65.79 (−319.34, 450.93) area under the curve (AUC), *p*-value = 0.610), N=3 WT + 4 ADAD donor lines, 1–3 independent differentiations per line). Further experiments are required to explain the mechanism underlying the apparent reduction of microglia–plaque association in vivo. For example, microglia activation, through immunostaining with specific markers, would be an interesting phenotype to investigate in the future.

### 2.5. Neuritic Pathology in AD Mice Xenotransplanted with WT or ADAD Human Microglia

A noted feature observed in the brains of individuals with AD and in amyloidogenic mouse models is the early presence of dystrophic neurites near amyloid plaques [32,33,34]. Dystrophic neurites are abnormal neuronal processes, such as swollen axons and dendrites, which substantially contribute to synaptic and neuronal pathology in AD. By staining the 5XFAD xenotransplanted mouse brains with AT8, an antibody for phosphorylated tau that accumulates in dystrophic neurites (Figure 4), we observed an increase in the number of dystrophic neurites (normalized by the plaque volume) in mice xenotransplanted with human ADAD microglia compared to WT human microglia (ADAD/WT (95% confidence interval) = 1.6 (1.04, 2.46), *p*-value = 0.055, N=3 WT + 4 ADAD donor lines, 1–3 xenotransplanted mice per donor line, 3 brain sections analyzed per mouse, 5 plaques quantified per section). These results may partially be explained by our earlier results demonstrating that (1) plaque compaction is reduced in the presence of ADAD human microglia, and (2) ADAD human microglia do not associate with plaques as closely as WT human microglia. These phenotypes together may be linked to an increase in dystrophic neurites.

### 2.6. Astrogliosis in AD Mice Xenotransplanted with WT or ADAD Human Microglia

Crosstalk between microglia and astrocytes has been widely reported [35], which led us to investigate astrocytic changes in our 5XFAD xenotransplanted mice using the marker GFAP. As shown in Figure 5, a non-statistically significant increase in the area covered by astrocytes (ADAD/WT (95% confidence interval) = 1.21 (0.67, 2.16), *p*-value = 0.531) and in GFAP intensity (ADAD/WT (95% confidence interval) = 1.18 (0.57, 2.08), *p*-value = 0.588), N = 3 WT + 4 ADAD donor lines, 1–3 xenotransplanted mice per donor line, 3 brain sections per mouse) was observed in 5XFAD mice xenotransplanted with ADAD compared to WT human microglia.

### 2.7. Behavior in AD Mice Xenotransplanted with WT or ADAD Human Microglia

Complementary to our characterization of morphological phenotypes, we investigated whether ADAD mutations in microglia might affect mouse behavior. *APP* and *PSEN1* mutations in mouse neurons have been shown to induce behavioral and cognitive deficits in different mouse models [26,36]. However, whether the expression of *APP* and/or *PSEN1* mutations exclusively in microglia is sufficient to cause these deficits remains largely unexplored. We first evaluated spatial working memory in 3-month-old 5XFAD mice xenotransplanted with either WT or ADAD microglia using the y-maze test of spontaneous alternation (Figure 6A). 5XFAD mice xenotransplanted with ADAD microglia exhibited a non-statistically significant impairment of working memory compared to 5XFAD mice xenotransplanted with WT microglia (ADAD-WT (95% confidence interval) = −5.53 (−18.35, 7.30) times, *p*-value = 0.387, N = 4 WT + 4 ADAD donor lines, 3–7 xenotransplanted mice per donor line). We repeated this task at 5 months of age (Figure 6B) and observed a similar non-statistically significant impairment of working memory in the 5XFAD mice xenotransplanted with ADAD microglia (ADAD-WT (95% confidence interval) = −11.83 (−31.11, 7.44), *p*-value = 0.252, N = 4 WT + 4 ADAD donor lines, 3–7 xenotransplanted mice per donor line). Notably, when we consider the decrease in working memory performance over time, the most significant decline in performance was within the 5XFAD group xenotransplanted with human microglia carrying *APP* mutations (Figure 6C).

We next investigated the performance of these same mice on contextual fear conditioning, a task in which AD model mice are known to exhibit behavioral deficits. Specifically, mice were placed in a training chamber and exposed to 4-foot shocks over the course of a 10-min training trial. 5XFAD mice xenotransplanted with ADAD microglia exhibited a non-statistically significant decrease in freezing in the 40 s following the final shock (ADAD-WT (95% confidence interval) = −11.28 (−24.94, 2.389) %, *p*-value = 0.157, N = 4 WT + 4 ADAD donor lines, 2–6 xenotransplanted mice per donor line), possibly indicating a decrease in learning acquisition over the training trial (Figure 6D). The next day, mice were placed back into the training chamber and the percentage of the test spent freezing was measured as an index of contextual fear memory. We found a decrease in percent freezing in 5XFAD mice xenotransplanted with ADAD microglia compared to WT microglia (ADAD-WT (95% confidence interval) = −11.05 (−22.25, 0.159) %, *p*-value = 0.101, N = 4 WT + 4 ADAD donor lines, 2–6 xenotransplanted mice per donor line), indicating a mild impairment in contextual memory.

In short, 5XFAD mice showed mild impairment in both contextual and spatial working memory when xenotransplanted with ADAD human microglia. These cognitive deficits were not due to gross differences in locomotor activity, as shown by no differences in total distance traveled (ADAD-WT (95% confidence interval) = −0.19 (−0.76, 0.39) cm, *p*-value = 0.546] or time in the center (ADAD-WT [95%confidence interval] = 36.32 (−50.23, 123.170) seconds, *p*-value = 0.400) in an open field arena (Appendix A).

## 3. Discussion

The role of ADAD mutations has been rarely studied in human microglial cells and, to our knowledge, our work is the first to study in vivo consequences of microglia-specific expression of these mutations. The xenotransplantation of human microglia into the mouse brain constitutes an innovative system to evaluate the effects of human genetic variants associated with AD in vivo. Here we showed that our technique leads to a 40–60% engraftment of human microglia using 8 different iPSC donor lines. The xenotransplantation protocol we utilized was based on a prior study, which demonstrated approximately 80% engraftment of human microglia [20]. However, their quantification was performed in a different mouse background (MITRG), at 2 months, in sagittal sections, and using only two different donor lines, which might account for this disparity. Using our same mouse strain but a different protocol, Fattorelli et al. reported a 60–80% engraftment in 2.5 month-old mice using 3 donor lines [37], a successful improvement from their previous work that showed a 9% human microglia engraftment using one donor line [38]. Future xenotransplantation studies are required to better characterize xenotransplantation variability across laboratories and donor lines.

We showed that *APP V717L*, *V717I, PSEN1 A79V*, and *G217R* mutations in microglial cells are not sufficient to cause amyloid pathology in the hCSF1 WT mouse brain. However, the xenotransplantation of human microglia carrying ADAD mutations in hCSF1-5XFAD mice showed a non-statistically significant increase in plaque volume and a reduction in plaque compaction. In accordance, mouse microglia containing phospho-deficient mutant PSEN1 have shown a reduced ability to degrade Aβ oligomers due to a phagolysosome dysfunction [39]. In human cortical neurons, *PSEN1* and *APP* mutations also lead to endosomal abnormalities [40]. However, due to the high variability and limited sample size, our results did not achieve statistical significance. These small effect sizes are in accordance with in vitro results from Konttinen et al., where they describe that *APP KM670/671NL* Swedish double mutation and *PSEN1 AE9* mutation in iPSC-derived microglia triggered minor alterations in microglia functionality (Konttinen et al. 2019). Future studies should use the results of our exploratory study to confirm the association between amyloid pathology and ADAD mutations in human microglia.

A reduction in microglia–plaque interaction has been reported in human microglia bearing other AD risk mutations, such as *TREM2-KO*, *TREM2-R47H* [30], or *PLCG2* variants [31]. In alignment with those studies, our findings revealed a decrease in the interaction between microglia and plaques, potentially contributing to the observed reduction in plaque compaction. We noted a decrease in the size of dense-cord plaques as indicated by ThioS staining, while pan-Aβ42 staining showed an increase in plaque volume, though these effects did not achieve statistical significance. The mechanisms underlying the potential reduction in microglia–plaque interactions downstream of ADAD or AD risk mutations need to be explored in further studies. A plausible hypothesis is that microglia containing ADAD mutations had a reduced motility capacity. Ledo et al. reported a slower kinetic response to laser-induced microinjury in phospho-deficient mutant *PSEN1* mouse microglia in vivo [39]. However, as Konttinen et al. have previously shown in *APPswe* and *PSEN1ΔE9* human microglia [18], we also observed a slight increase in the motility capacity of human microglia when bearing ADAD mutations. In parallel, McQuade et al. showed that *TREM2-KO* microglia also reduced microglial plaque association in vivo without displaying any alterations in motility in in vitro experiments [41].

Mabrouk et al. have recently demonstrated that most dystrophic neurites in 5XFAD mice result from Aβ accumulation in axon terminals [42]. Alternatively, presynaptic dystrophic neurites surrounding amyloid plaques have been identified as sites of additional Aꞵ generation [43]. Together, these findings suggest an important link between Aꞵ accumulation and/or generation and dystrophic neurites. Here, we observed an increase in the number of dystrophic neurites in the 5XFAD mice xenotransplanted with human microglia bearing ADAD mutations compared to 5XFAD mice xenotransplanted with WT human microglia. When combined with our reported increase in Aβ plaque volume in 5XFAD mice xenotransplanted with ADAD microglia compared to 5XFAD mice xenotransplanted with WT microglia, our results support this overall hypothesis, although underlying mechanisms remain to be investigated. On the other hand, it has been shown that the regions surrounding plaques that are covered by microglial processes develop less dystrophic neurites [44]. Therefore, the observed reduction in plaque-associated microglia in 5XFAD xenotransplanted with ADAD microglia could contribute to the increased number of dystrophic neurites. Finally, Konttinen et al. have shown that iPSC-derived microglia expressing ADAD mutations exhibit impaired responsiveness to ATP compared to WT iPSC-derived microglia [18], the recognition of which is the initiating step in a cascade of events for the suppression of neuronal activity [45]. An alteration in microglia–neuronal interaction may also contribute to the increase in the neuritic pathology observed in vivo.

In addition to their role in modifying neuropathological phenotypes, microglia have also been reported to significantly contribute to the dynamic processes involved in learning and memory. The role that microglia play in these processes has been well-documented to change depending on activation state, local and systemic inflammation, as well as aging and disease [46]. Therefore, we hypothesized that the expression of ADAD mutations in microglia may modify the progression of cognitive decline in 5XFAD mice. In support of this hypothesis, we observed modest though non-statistically significant deficits in both contextual fear learning and memory in 5XFAD mice xenotransplanted with ADAD microglia relative to those transplanted with WT microglia. In addition, we observed similar deficits in working memory across groups at both 3 and 5 months of age. Notably, 5XFAD mice xenotransplanted with *APP* ADAD microglia exhibited the most significant working memory decline across this time period. This unique difference across *APP* and *PSEN1* ADAD microglia was not observed in the morphological phenotypes measured, suggesting that *APP* ADAD mutations may have an effect on cognitive outcomes independently of the degree of neuropathology (i.e., plaque morphology, load, or presence of dystrophic neurites). Given the complex relationship between neuropathology and cognitive outcomes, this represents an intriguing new alternative mechanism for *APP* in AD that should be explored in future studies.

The moderate size and high variability of effects estimated in our study coupled with our limited sample size prevented most results from reaching statistical significance. However, this work will be of value to future studies using human microglia derived from population iPSC lines, which may include the same or additional mutations. For example, to achieve statistical significance in the increment of dystrophic neurites in 5XFAD mice xenotransplanted with ADAD microglia compared to those xenotransplanted with WT microglia, an estimated sample size of eight donor lines per genotype will be required for 80% power, more than what is currently available through projects such as DIAN. Another caveat of this study is the lack of isogenic lines to be used as a control, due to their unavailability at the beginning of the study. However, for *APP V717I, PSEN1 A79V,* and *PSEN1 G217R* mutations, we have non-carrier donor lines from members of the same family, which might mitigate this limitation. As *APOE* is the main non-autosomal dominant AD risk gene, the *APOE* genotype is shown in Table 1. *APOE* ε3 is the most common allele and is considered neutral in terms of AD risk, while *APOE* ε2 is associated with a reduced risk of developing the disease, and *APOE* ε4 is linked to an increased risk, compared to *APOE* ε3. Although the use of population lines is more representative of human diversity, the generation of isogenic lines and further xenotransplantation in mice will be necessary to undoubtedly establish the causal effect of ADAD mutations on microglial function. A gene-corrected human isogenic iPSC line from the *APP V717I* line is already available [47] and *PSEN1* isogenic lines are being generated (unpublished data).

While our work specifically investigated autosomal dominant ADAD-causing mutations, AD is a multifactorial disorder with a complex interplay of genetic and environmental factors. Therefore, focusing on two genes may not adequately capture the heterogeneity observed in the broader Alzheimer’s patient population. The simultaneous xenotransplantation of human microglia from donors with high and low polygenic risk scores for AD or different APOE genotypes in AD mouse models would better recapitulate this diversity. To pilot this approach the co-culture of microglia derived from multiple donor lines is being optimized (‘village’ culture system) in vitro and in vivo.

## 4. Materials and Methods

### 4.1. Mouse Models Used

Non-transgenic (wild-type, WT) and 5XFAD (Forner et al. 2021 [48]) immune-deficient mice (hCSF1) capable of accepting human cells were utilized. hCSF1 mice (available as JAX #017708) express human mCSF (CSF1) on a Rag2/Il2rg deficient background, significantly increasing their ability to engraft and maintain human hematopoietic cells. hCSF1-5XFAD mice overexpress co-integrated APP and PSEN1 transgenes containing five familial AD mutations (the Swedish (*K670N/M671L*), Florida (*I716V*), and London (*V717I*) mutations in *APP*, and the *M146L* and *L286V* mutations in *PSEN1*) and recapitulate aspects of human AD, including age-dependent cognitive deficits, amyloid accumulation and neuroinflammation [25]. The offspring were backcrossed to hCSF1 mice to return the 3 hCSF1 mutations to homozygosity and maintain the 5XFAD transgene in the hemizygous state [20]. These 5XFAD-hCSF1 mice were maintained in the Goate laboratory. All procedures involving experimentation on animal subjects were done in accord with the National Research Council’s guide for the care and use of laboratory animals and under an approved protocol.

### 4.2. Induced Pluripotent Stem Cells (iPSC) Donor Lines and Hematopoietic Progenitor Cells (HPC) Generation

iPSCs were obtained from the iPSC repository of the DIAN project, directed by Dr. Karch (Washington University, St. Louis, MO, USA) [49]. Donor line information is detailed in Table 1. iPSC lines were thawed into Matrigel-coated 6-well plates and maintained by feeding every other day with Stemflex supplemented media (Thermo Fisher Scientific, Waltham, MA, US) at 37 °C, 5% CO_2_. When 70–80% confluence was reached, iPSCs were passaged using ReLeSR dissociation reagent (#05872, STEMCELL Technologies, Vancouver, Canada). Hematopoietic stem cells were differentiated from iPSCs using the STEMdiff Hematopoietic kit (#05310 STEMCELL Technologies, Vancouver, Canada,) according to a previously published protocol [41]. When cultured for a minimum of 10 days, HPCs were collected and, if viability was >75%, frozen in Bambanker media (Wako Chemicals, Osakan, Japan).

### 4.3. Differentiation to iPSC-Derived Microglia and Scratch Wound Assay

Hematopoietic stem cells were differentiated to induced microglial-like cells (iMGLs) using a previously published protocol [41]. iMGLs were maintained and fed with a microglial medium supplemented with three factors (100 ng/mL IL34, 50 ng/mL TGFβ, 25 ng/mL MCSF) for 25 days. On day 25, iMGLs were additionally supplemented with two factors (CX3CL1 and CD200, 100 ng/mL each) for an additional three days. Mature iMGLs (day 28) were used for the Incucyte Scratch Wound Assay (Sartorius, Gottingen, Germany), a real-time, automated method for studying cell migration. On day 25, mature iMGLs (30,000 cells per well) were seeded in a 96-well tissue culture plate. On day 28, a scratch/wound was generated using a 96-pin WoundMaker tool. Two washes were performed to eliminate cell debris. The plate was then placed into the Incucyte Live-Cell Imaging System, where images were captured every hour. The Incucyte software (2022B Rev3 version) analyzed the images and enabled the quantification of wound closure over time. We performed 1–3 independent experiments per line, with 4–8 technical replicates per experiment that were averaged within each experiment.

### 4.4. Xenotransplantation

iPSC-derived hematopoietic stem cells (HPCs) from lines either harboring *APP* or *PSEN1* FAD mutations or non-carrier controls were injected into the hippocampus and cortex of neonatal wild-type and 5XFAD hCSF1 mice, according to an established protocol [20]. P2 pups from the transplantation-competent hCSF1 mouse strain were placed on ice to induce hypothermic anesthesia. The site of the injection was disinfected. In between donor HPCs, needles were cleaned with consecutive washes of PBS, 70% ethanol, followed by PBS. Tools were kept sterile. Free-hand transplantation was performed using a 30-gauge needle affixed to a 10 μL Hamilton syringe. Mice received 1 μL of HPCs suspended in 1x DBPS at 62.5 K cells/μL at each of 8 injection sites, totaling 500 K cells/pup. Bilateral injections were performed at 2/5th of the distance from the lambda suture to each eye, injecting into the lateral ventricles at 3 mm and into the overlying anterior cortex at 1 mm, and into the posterior cortex in line with the forebrain injection sites, and perpendicular to lambda at a 45° angle. Transplanted pups were then returned to their home cages and weaned at P21.

### 4.5. Behavior Tests

Prior to all behavior tests, mice were habituated to transfer for at least three days. Tests were carried out from least to most stressful in the following order: y-maze, open field, and contextual fear memory. To assess spatial working memory, the y-maze test of spontaneous alternation was performed, as described previously [25,36]. The y-maze used for testing was made of clear acrylic with three arms each at a 120-degree angle from the other. The maze was placed on a table in a dimly lit room and spatial cues were displayed on walls around the table. Mice were placed in a start arm and Ethovision video tracking software (version XT14) was used to monitor arm entries.

On the following day, we conducted the open field test by introducing the mouse to a new, well-lit arena and permitting it to freely explore for one hour. By using the Fusion software (version v.5.0), the arena automatically recorded various behaviors, including the total distance covered. The arena was manually divided into ‘center’ and ‘periphery’ zones and the time spent, as well as distance traveled, in each of these zones was measured.

After a minimum of one week following the above behavioral tests, mice underwent contextual fear conditioning according to previously described protocols [36]. The initial training trial consisted of a 180 s baseline period followed by randomly spaced 4-foot shocks (1 s, 0.5 mA). Mice were then returned to their home cage for 24 h. The next day, the mice were returned to the same chamber for a 10-min testing trial. On both training and testing days, the amount of time the mouse spent freezing was measured using VideoFreezeTM software (version v.2.7.3).

### 4.6. Immunohistochemistry (IHC)

Six-month-old mice were perfused with PBS and a hemibrain was fixed in PFA 4%, cryoprotected in sucrose solution and stored at −80 °C. Hemibrains were coronally sectioned in a cryostat (30–40 μm) and stored as free-floating sections in a cryoprotective solution at −20 °C. For IHC, 4–8 sections per mouse were selected and placed on a 24-well dish (maximum of 4 sections per well). Tissue sections were washed three times in TBS-Tween (TBST) buffer for 5 min each, followed by a 20-min incubation in pretreatment buffer (0.6% H_2_O_2_, 0.1% Triton-X in TBST). Sections were rewashed in TBST buffer three times and transferred to Eppendorf tubes to be treated with sodium citrate pH 6 buffer on a heat block at 80 °C for 30 min. Next, sections were transferred back to the 24-well plate, washed three times in TBST, and incubated overnight at 4 °C in blocking solution (5% normal donkey serum in TBST + 0.3% Triton X100). The following day, sections were incubated in primary antibodies Ku80 (Cell Signaling Technologies (Danvers, MA, USA) #2180, 1:100), IBA1 (Abcam (Cambridge, UK), ab5076 1:200), GFAP (Abcam ab5076, 1:1000), AT8 (Thermo Scientific (Waltham, MA, USA), MN1020B) 1:250) for 2 h at 37 °C. After three washes with TBST buffer, sections were incubated for 1.5 h at room temperature with secondary antibodies (Thermo Scientific 1:200) diluted in TBST. For additional staining for amyloid pathology, sections were washed again in TBST buffer three times and incubated with the antibody Aβ42 (β-Amyloid clone D54D2 Alexa Fluor 488 Conjugate, Cell Signaling Technologies #51374S, 1:250) overnight; or with Thioflavin S (Sigma-Aldrich (St. Louis, MO, USA) #T1892, 1:10,000) for 2 min. In all cases, sections were washed again and stained with DAPI (Thermo Fisher #D1306, 300 μM) for 5 min at room temperature. Sections were washed three times and mounted using PBS onto superfrost slides with TrueBlackmounting media (Biotium, Fremont, CA, USA). A coverslip was then slowly lowered onto each section, and the slides were stored at room temperature for 5–6 h before long-term storage at 4 °C.

### 4.7. Image Analysis

For total microglia number, percentage of human microglia, amyloid quantifications, and total astrocyte number, a Keyence microscope was used to image complete brain sections (single-plane, 10X, stitched images). For the microglia and amyloid beta analysis, a custom ImageJ macro was used. First, the DAPI channel image was used to define the region of interest (cortex and hippocampus). Then the macro opened the corresponding Aβ (Aβ42 or Thioflavin S)- or microglia (BA1 or Ku80)-stained image. We established a threshold and quantified the fluorescence integrated density, the percentage of area stained, and the number of stained objects (plaques, cells). To analyze engraftment efficiency, the number of human microglia (Ku80+ cells) was divided by the total microglia number (IBA1+ cells).

To study the number of plaque-associated microglia and area covered, plaque morphology and volume, and the number of dystrophic neurites, confocal microscopy was used (z-stack 20–25 slices, 40X, tile 2 × 3) and Imaris software (version 10.0) was used for 3D reconstruction. Volume masks were generated for microglia and plaques using a Gaussian filter and specific thresholds. The number of microglia associated with amyloid plaques was manually counted and divided by the corresponding plaque volume. To calculate the area covered by microglia, a 5 μm extension was generated from the perimeter of the plaque, and the overlap between amyloid and microglia channels was measured. For dystrophic neurites analysis, the maximum intensity projection was generated from the z-stack images in FIJI, and AT8+ neurites closer than 5 μm from the border of the plaque were manually counted and divided by the corresponding plaque size.

### 4.8. Statistical Analysis

Image analysis was performed blinded to the HPC xenotransplanted genotype. R studio software 2021.09.0 was used to visualize and analyze the data (R Core Team (2023). A Language and Environment for Statistical Computing. R Foundation for Statistical Computing, Vienna, Austria; https://www.R-project.org/, accessed on 11 December 2023). Differences of means between groups were tested with lmer function [50]. Mouse sex and cell line mutations were added as covariates in the statistical analysis. As the human microglia engraftment varied across mice, we also included the percentage of human microglia as a covariate in the analysis (except for the analysis of the behavioral tests). When data were normalized by the plaque size/volume (such as for plaque-associated microglia and dystrophic neurites measurements), plaque size or volume was calculated as ‘plaque volume/average of all plaque volumes’. Raw data files and R scripts are available as Appendix A (Raw data and analysis) for further details.

## 5. Conclusions

In short, although ADAD mutations in human microglia are not sufficient to initiate amyloid deposition in the brain of WT mice, they cause mild changes in AD pathology in the brain of an AD mouse model, specifically altering amyloid plaque size, microglia-plaque interaction, and neuritic pathology. In addition, we observed modest deficits in both contextual fear learning and memory as well as in working memory in 5XFAD mice xenotransplanted with ADAD microglia relative to mice transplanted with WT microglia. Further work is required to confirm these effects and better understand the mechanisms underlying these alterations.

## Figures and Tables

**Figure 1 ijms-25-02565-f001:**
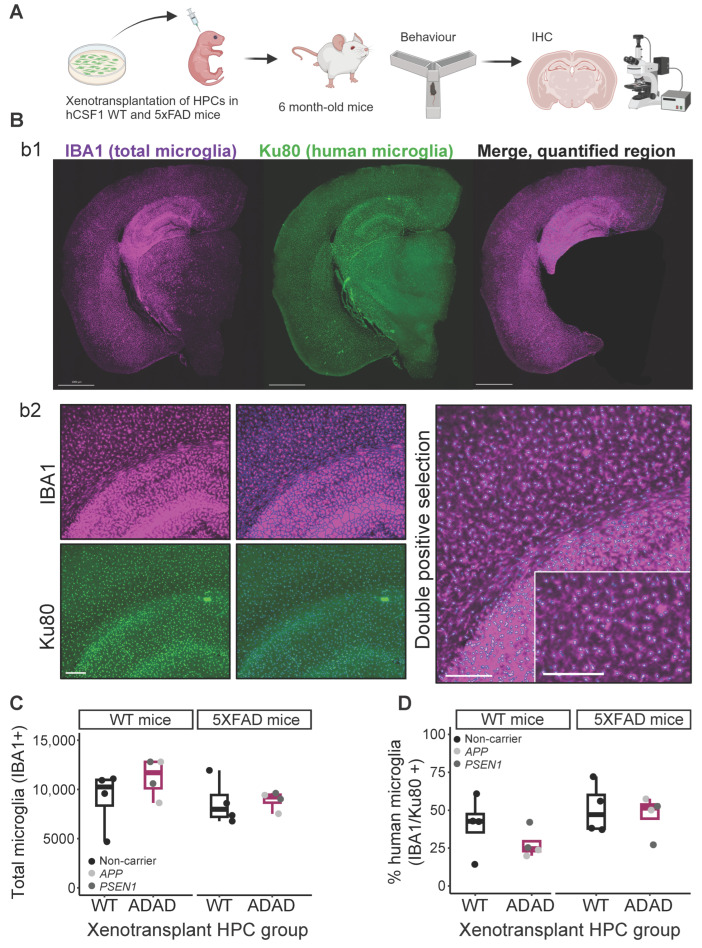
Microglial engraftment in both WT and 5XFAD immune-deficient hCSF1 mice. (**A**) Experimental design. Hematopoietic progenitor cells (HPCs) were generated from induced pluripotent stem cells (iPSC) from different donor lines (see Table 1) and xenotransplanted in immunocompromised (hCSF1) WT or 5XFAD mice. At 3–6 months old, mice were subjected to behavioral tests, then sacrificed at 6 months old. Brain sections were subjected to immunohistochemistry analysis. (**B**) Representative images for total (IBA1) and human (Ku80) microglia in the mouse brain: b1, brain section, scale bar = 1000 μm; b2, representative image of microglia segmentation for quantification in ImageJ software 1.54f version, scale bar = 200 μm. (**C**) Quantification of total number of microglia (IBA1+ cells), and (**D**) percentage of human microglia (Ku80+/IBA1+*100) in the cortex and hippocampus in 6-monthold hCSF1 WT and 5XFAD mice xenotransplanted with WT or ADAD HPCs. Each dot represents one HPC donor line (N = 4 WT + 4 ADAD donor lines, 1–5 xenotransplanted mice per donor line, 4–7 brain sections analyzed per mouse).

**Figure 2 ijms-25-02565-f002:**
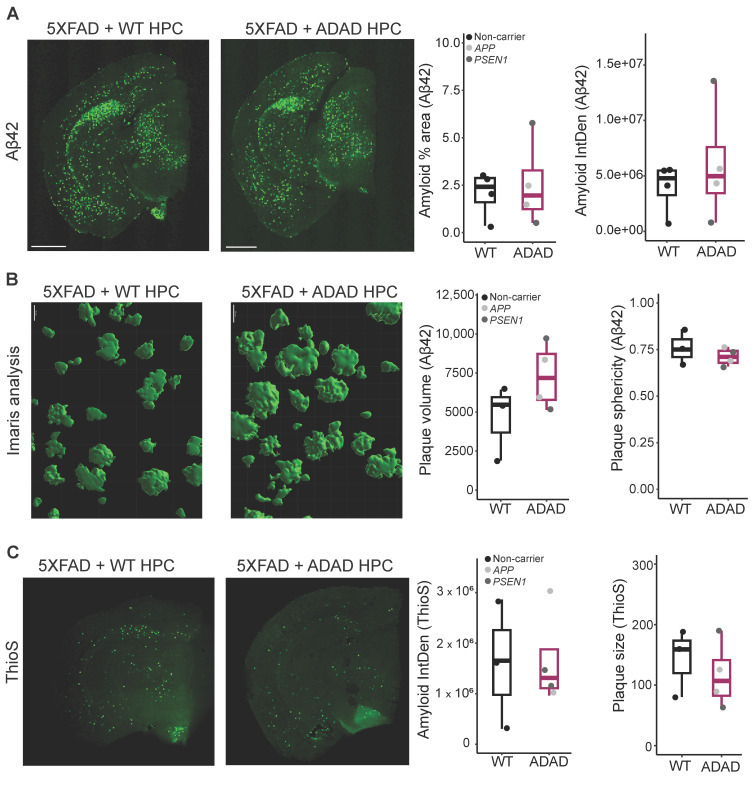
(**A**) Representative images for Aβ42 amyloid staining, scale bar = 1000 μm. Quantification of Aβ42 staining: amyloid percentage area and amyloid integrated density in cortex and hippocampus of 6-month-old hCSF1 WT and 5XFAD mice xenotransplanted with WT and ADAD HPCs. Each dot represents one HPC donor line (N = 4 WT + 4 ADAD donor lines, 1–5 xenotransplanted mice per donor line, 4–7 brain sections analyzed per mouse). (**B**) Aβ42 staining, quantification of plaque volume and plaque sphericity, using reconstruction in Imaris software (version 10.0), (scale bar = 20 μm), in the cortex of 6-month-old hCSF1 WT and 5XFAD mice xenotransplanted with WT and ADAD HPCs. Each dot represents one HPC donor line (N = 3 WT + 4 ADAD donor lines, 1–3 xenotransplanted mice per donor line, 3 brain sections analyzed per mouse, 5 plaques quantified per section). (**C**) Representative images for ThioS staining, scale bar = 1000 μm. Quantification of dense core plaques (ThioS staining): amyloid integrated density, and average plaque size in cortex plus hippocampus of 6-month-old hCSF1 WT and 5XFAD mice xenotransplanted with WT or ADAD HPCs. Each dot represents one HPC donor line (N = 3 WT + 4 ADAD donor lines, 1–3 xenotransplanted mice per donor line, 4–7 brain sections analyzed per mouse).

**Figure 3 ijms-25-02565-f003:**
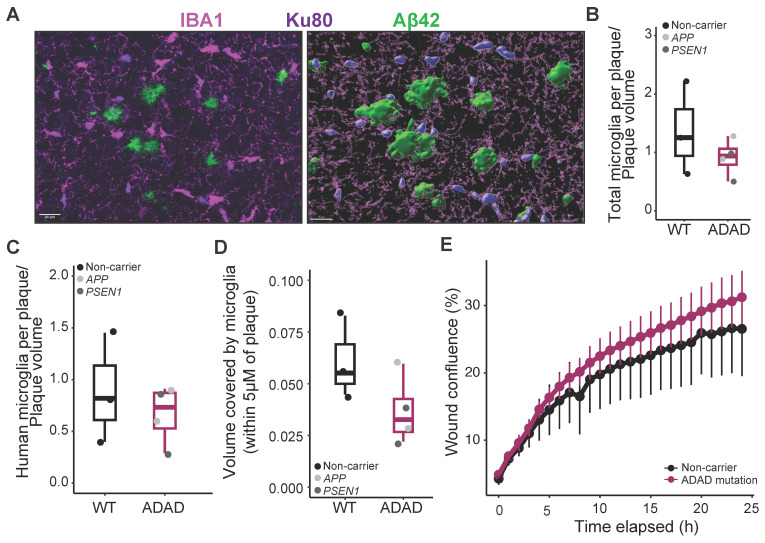
Characterization of microglia association to amyloid plaques. (**A**) Representative image of total (IBA1) and human (Ku80) microglia associated with amyloid plaques (Aβ42), and example of surface generation for quantification in Imaris software (version 10.0). Scale bar = 20 μm. Quantification of total (**B**) or human (**C**) number of microglial cells in direct contact with the plaque, normalized by the plaque volume. (**D**) Area covered by microglia in the plaque and surroundings (including a ring of 5 μm from the border of the plaque) in the cortex of 6-month-old hCSF1 5XFAD mice xenotransplanted with WT and ADAD HPCs. Each dot represents one donor line (N = 3 WT + 4 ADAD donor lines, 1–3 xenotransplanted mice per donor line, 3 brain sections analyzed per mouse, 5 plaques quantified per section). (**E**) In vitro migration assay (scratch wound confluence) in WT, and ADAD iPSC-derived microglia (N = 3 WT + 4 ADAD donor lines, 1–3 independent differentiations per line).

**Figure 4 ijms-25-02565-f004:**
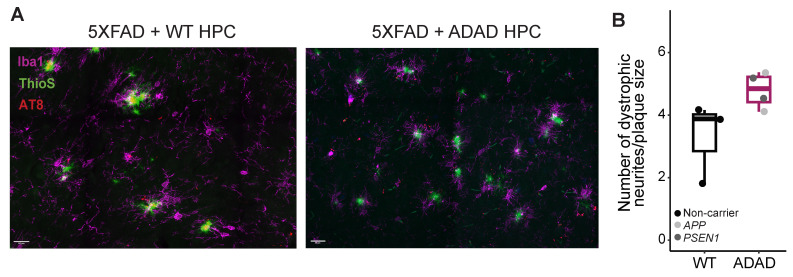
Representative images, scale bar =20 μm (**A**) and quantification (**B**) of dystrophic neurites (AT8+) in the cerebral cortex of 6-month-old hCSF1 5XFAD mice xenotransplanted with WT and ADAD human microglia. Each dot represents one donor line (N = 3 WT + 4 ADAD donor lines, 1–3 xenotransplanted mice per donor line, 3 brain sections analyzed per mouse, 5 plaques quantified per section).

**Figure 5 ijms-25-02565-f005:**
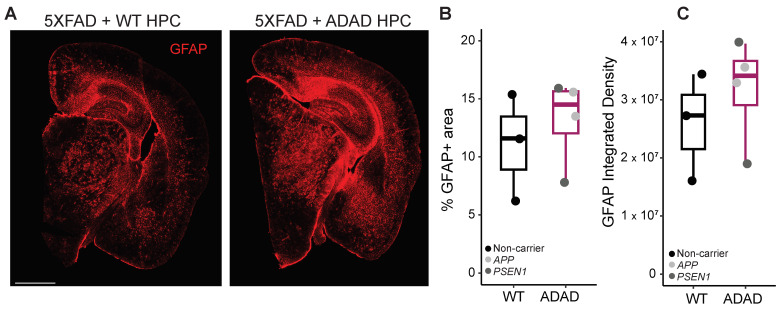
Quantification of astrocytes (GFAP+ staining) in the cortex and hippocampus of 6-month-old hCSF1 5XFAD mice xenotransplanted with WT and ADAD HPCs. (**A**) Representative image. Scale bar = 1000 μm. Quantification of percentage area covered by astrocytes (**B**) and GFAP integrated density (**C**). Each dot represents one donor line (N = 3 WT + 4 ADAD donor lines, 1–3 xenotransplanted mice per donor line, 3 brain sections analyzed per mouse).

**Figure 6 ijms-25-02565-f006:**
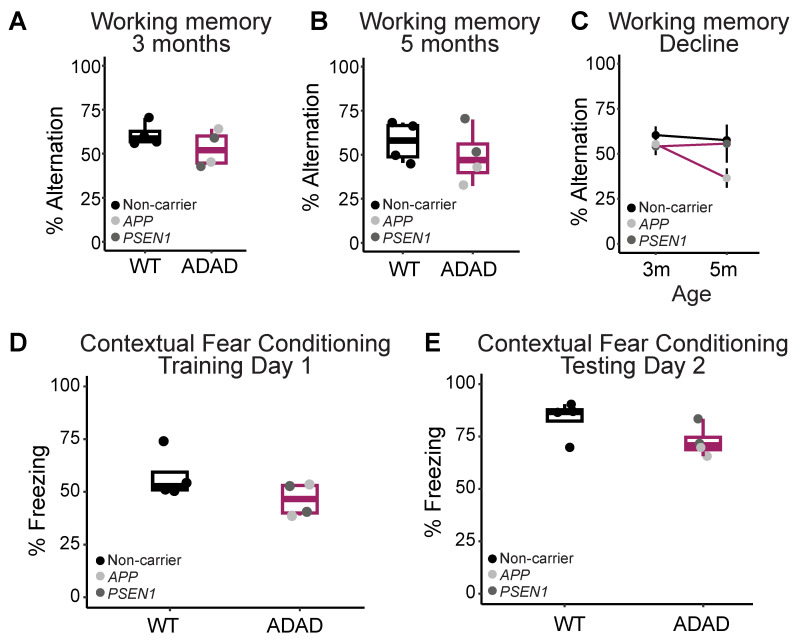
Behavior tests to evaluate cognitive function. (**A**) Working memory in 3 and (**B**) 5-month-old hCSF1 WT and 5XFAD mice xenotransplanted with WT and ADAD HPCs. (**C**) 5XFAD mice xenotransplanted with *APP* ADAD human microglia exhibit the most significant working memory decline from 3 to 5 months. (**D**) Contextual fear acquisition in 6-month-old 5XFAD mice as quantified by the percentage of time spent freezing during the 40 s following the final shock in the training trial. (**E**) Contextual fear memory in 6-month-old hCSF1 WT and 5XFAD mice xenotransplanted with WT and ADAD HPCs as quantified by the percentage of the 10-min testing trial spent freezing. Each dot represents one donor cell line (N = 3–4 WT + 4 ADAD donor lines, 2–7 xenotransplanted mice per donor line) except for (**C**) where one dot represents the average for all mice of a given xenotransplant genotype.

**Table 1 ijms-25-02565-t001:** Information on the induced-pluripotent stem cells (iPSC) lines used. All donor lines are Caucasian. *APOE* genotype is shown in the table because it is the main non-autosomal dominant AD risk gene. *APOE* ε3 is the most common allele and is considered neutral in terms of AD risk, while *APOE* ε2 is associated with a reduced risk of developing the disease, and *APOE* ε4 is linked to an increased risk, compared to *APOE* ε3.

iPSC Line	Genotype	Sex	APOE Genotype
F12455	APP and PSEN1 mutation Negative	F	3/3
F15553.3	APP V717L Positive	M	3/3
F16574.1	APP V717I Positive	M	3/3
FA12462	APP V717I Negative	M	3/3
F12424.2	PSEN1 A79V Positive	M	3/4
F12436.2	PSEN1 A79V Negative	M	3/3
F12434.1	PSEN1 G217R Positive	M	2/4
F12445.5	PSEN1 G217R Negative	F	3/4

## Data Availability

The authors enclosed a folder with all raw data and R analysis. Files are named regarding the corresponding subsections in the Section 2.

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
