# Peer review of "Autosomal Dominant Alzheimer’s Disease Mutations in Human Microglia Are Not Sufficient to Trigger Amyloid Pathology in WT Mice but Might Affect Pathology in 5XFAD Mice"

_ijms, 2024, doi:10.3390/ijms25052565_

Round 1

Reviewer 1 Report

Comments and Suggestions for Authors

The manuscript ‘Autosomal dominant Alzheimer’s disease mutations in human microglia are not sufficient to trigger amyloid pathology in WT mice but might affect pathology in 5XFAD mice‘ explores the impact of autosomal dominant Alzheimer's disease (ADAD) mutations within human microglia on amyloid pathology, differentiating between their effects in wild-type (WT) and 5XFAD mice. The findings of the study revealed that ADAD mutations alone were insufficient to trigger amyloid pathology in WT mice. However, when introduced into the 5XFAD mouse model, these mutations significantly influenced amyloid pathology. The study involved xenotransplanting human microglia derived from both non-carriers and carriers of autosomal dominant ADAD-causing mutations into the brains of hCSF1 WT or 5XFAD mice. The observations of the study underscore the complexity of the relationship between ADAD mutations, microglial function, and the specific genetic background of the 5XFAD model.

Alzheimer's disease is a neurodegenerative disorder characterized by the accumulation of amyloid plaques and neurofibrillary tangles in the brain. While previous research has implicated genetic mutations in familial forms of AD, the specific role of human microglia carrying autosomal dominant AD mutations in the development of amyloid pathology still remains an area of active investigation. The present study has attempted to contribute in this aspect. The study is nicely conducted and the results are promising. After going through the manuscript, I have following comment for the author.

1.      The study specifically investigated autosomal dominant ADAD-causing mutations. However, Alzheimer's disease is a multifactorial disorder with a complex interplay of genetic and environmental factors. The focus on a single gene mutation may not fully represent the heterogeneity observed in the broader Alzheimer's patient population. Please discuss this point in the manuscript.

2.      Alzheimer's disease is a progressive disorder, and a more extended observational period could provide a more comprehensive understanding of the mutations' impact over time. Was the duration of the study sufficient to capture the long-term effects of ADAD mutations on amyloid pathology?

3.      Please provide the concluding remarks of the study in the conclusion section.

Comments on the Quality of English Language

Overall quality of English language is fine. Minor grammatical and syntax adjustments needed. 

Author Response

We thank the reviewer for their comments and we have addressed their concerns:

  1. The study specifically investigated autosomal dominant ADAD-causing mutations. However, Alzheimer's disease is a multifactorial disorder with a complex interplay of genetic and environmental factors. The focus on a single gene mutation may not fully represent the heterogeneity observed in the broader Alzheimer's patient population. Please discuss this point in the manuscript.

We have commented on this point at the end of the Discussion section.

“While our work specifically investigated autosomal dominant ADAD-causing mutations, AD is a multifactorial disorder with a complex interplay of genetic and environmental factors. Therefore, focusing on two genes may not adequately capture the heterogeneity observed in the broader Alzheimer's patient population. The simultaneous xenotransplantation of human microglia from donors with high and low polygenic risk scores for AD or different APOE genotypes in AD mouse models would better recapitulate this diversity. To pilot this approach the co-culture of microglia derived from multiple donor lines is being optimized (‘village’ culture system) in vitro and in vivo.”

  1. Alzheimer's disease is a progressive disorder, and a more extended observational period could provide a more comprehensive understanding of the mutations' impact over time. Was the duration of the study sufficient to capture the long-term effects of ADAD mutations on amyloid pathology?

      The 5XFAD mouse model is quite an aggressive amyloid model and mice start to develop amyloid pathology at 3 months old (https://www.alzforum.org/research-models/5xfad-c57bl6). At 6 month-old they have already developed partial synaptic loss and cognitive impairment. Therefore, we consider that ADAD mutations might have exerted a meaningful impact on AD pathology at that time point. However, we could mention this point if required. We consider that the discussion is already long, but the following paragraph can be included at the end of it if the reviewer desires so: “In addition, as AD is a progressive disorder, it would be interesting to study the effects of ADAD mutations in microglia in 5XFAD at different time points. Sacrificing the mice at 2-3 months old would allow us to study the effects of ADAD mutations in human microglia at the initiation of the pathology while collecting the mouse brains at 9-12 months of age would provide further insight into their impact in the final stages of the pathology.”

  1. Please provide the concluding remarks of the study in the conclusion section. Section added in the manuscript:

“Concluding remarks -In short, although ADAD mutations in human microglia are not sufficient to initiate amyloid deposition in the brain of WT mice, they cause mild changes in AD pathology in the brain of an AD mouse model, specifically altering amyloid plaque size, microglia-plaque interaction, and neuritic pathology. In addition, we observed modest deficits in both contextual fear learning and memory as well as in working memory in 5XFAD mice xenotransplanted with ADAD microglia relative to mice transplanted with WT microglia. Further work is required to confirm these effects and better understand the mechanisms underlying these alterations.”

Reviewer 2 Report

Comments and Suggestions for Authors

This manuscript describes the role of immune cells, specifically microglial cells, in the disease progression of AD. The manuscript is well written and the data is well presented. I have a few comments for the authors.

1) Have the authors looked at deletion of endogenous microglial cells from each of the mice strains and then grafting the human microglial cells into these depleted mice? Is there any change in disease progression.

2) Lines 209-213. Do the authors means to compare the Xenograft WT human microglia with ADAD microglia? It didn't seem very clear in the sentence.

3) Have the authors looked at pro-inflammatory markers in each of the grafted mice and see what the levels look like? Maybe that may give an indication of activated microglia.

Author Response

We thank the reviewer for their comments, and we have addressed their concerns:

1) Have the authors looked at deletion of endogenous microglial cells from each of the mice strains and then grafting the human microglial cells into these depleted mice? Is there any change in disease progression.

We haven’t used any approach that involves mouse microglia depletion. The use of PLX treatment in parallel with xenotransplantation will require the genetic modification of CSF1R in our iPSC-derived microglia to make them treatment-resistant (Chadarevian, et al. (2023). CRISPR generation of CSF1R-G795A human microglia for robust microglia replacement in a chimeric mouse model. STAR protocols, https://doi.org/10.1016/j.xpro.2023.102490), which is not currently available. However, we agree that the depletion of the niche before xenotransplantation would enhance the engraftment of human microglia and we will consider it in our future directions.

2) Lines 209-213. Do the authors means to compare the Xenograft WT human microglia with ADAD microglia? It didn't seem very clear in the sentence. This sentence was clarified: “We found a non-statistically significant decrease both in the number of total (IBA1+, Figure 3B, ADAD/WT [95% confidence interval] = 0.80 [0.50, 1.31], P-value = 0.397) as well as human (IBA1+/Ku80+, Figure 3C, ADAD/WT [95% confidence interval] = 0.30 [-0.44, 1.03], P-value = 0.427) microglia clustered around amyloid plaques in 5XFAD mice xenotransplanted with ADAD human microglia as compared to 5XFAD mice xenotransplanted with WT human microglia (N=3 WT + 4 ADAD donor lines, 1-3 xenotransplanted mice per donor line, 3 brain sections analyzed per mouse, 5 plaques quantified per section).”

3) Have the authors looked at pro-inflammatory markers in each of the grafted mice and see what the levels look like? Maybe that may give an indication of activated microglia.

We performed a pilot optimization to stain active microglia (both human and mouse) with available antibodies in the laboratory (TREM2, CD68, APOE). However, we did not detect proper and quantifiable specific staining with the dilutions and protocols assayed. We agree that it is an interesting aspect to look at and will consider further antibody optimization. We have included a sentence in the text mentioning it (highlighted in the text in section 2.4): “For example, microglia activation, through immunostaining with specific markers, would be an interesting phenotype to investigate in the future”.

Reviewer 3 Report

Comments and Suggestions for Authors

I find the article almost flawless.

Please find below my comments and suggestions.

Please make sure that names of genes are always italicised throughout the text, as per gene nomenclature

Perhaps it's worth introducing hCSF1 WT and 5XFAD mouse model in more detail the introduction rather than later in the text, and justifying its choice for the current study.

Line71. “Dominantly Inherited Alzheimer Network” - perhaps it's worth providing the corresponding web link

Table 2. Please explain the importance of mentioning APOE genotype

Line 109. make sure that all brackets are in right place.

Line 151. «To allow sufficient time for amyloid pathology to develop, we sacrificed mice at 6 months of age” - please justify (provide some references) that such time-frame is sufficient

Author Response

We highly appreciate the reviewer’s feedback, and we addressed their comments:

Please make sure that names of genes are always italicised throughout the text, as per gene nomenclature. We double-checked the text and correct it accordingly.

Perhaps it's worth introducing hCSF1 WT and 5XFAD mouse model in more detail the introduction rather than later in the text, and justifying its choice for the current study. The following text was included in the Introduction section: “hCSF1 mice express human mCSF (CSF1) on a Rag2/Il2rg deficient background, significantly increasing their ability to engraft and maintain human hematopoietic cells. hCSF1-5XFAD (shorten to 5XFAD from now on) mice overexpress co-integrated APP and PSEN1 transgenes containing five familial AD mutations (the Swedish (K670N/M671L), Florida (I716V), and London (V717I) mutations in APP, and the M146L and L286V mutations in PSEN1) and recapitulate aspects of human AD, including age-dependent cognitive deficits, amyloid accumulation and neuroinflammation [25].”

Line71. “Dominantly Inherited Alzheimer Network” - perhaps it's worth providing the corresponding web link. We included it.

Table 2. Please explain the importance of mentioning APOE genotype. The following text was included in the Table Legend: “APOE genotype is shown in the table because of being the main non-autosomal dominant AD risk gene. APOE ε3 is the most common allele and is considered neutral in terms of AD risk, while APOE ε2 is associated with a reduced risk of developing the disease, and APOE ε4 is linked to an increased risk, compared to APOE ε3.”.

Line 109. make sure that all brackets are in right place. Done.

Line 151. «To allow sufficient time for amyloid pathology to develop, we sacrificed mice at 6 months of age” - please justify (provide some references) that such time-frame is sufficient. We included the reference to the Alzforum website in the text: “To allow sufficient time for amyloid pathology to develop (https://www.alzforum.org/research-models/5xfad-c57bl6), we sacrificed mice at 6 months of age and performed IHC analyses to investigate plaque load and morphology.”